# The energy cost of polypeptide knot formation and its folding consequences

Andrés Bustamante[1], Juan Sotelo-Campos[2], Daniel G. Guerra[3], Martin Floor[1], Christian A.M. Wilson[1], Carlos Bustamante[3,4] & Mauricio Báez[1]

Knots are natural topologies of chains. Yet, little is known about spontaneous knot formation in a polypeptide chain—an event that can potentially impair its folding—and about the effect of a knot on the stability and folding kinetics of a protein. Here we used optical tweezers to show that the free energy cost to form a trefoil knot in the denatured state of a polypeptide chain of 120 residues is $5.8 \pm 1 \, \text{kcal mol}^{-1}$. Monte Carlo dynamics of random chains predict this value, indicating that the free energy cost of knot formation is of entropic origin. This cost is predicted to remain above $3 \, \text{kcal mol}^{-1}$ for denatured proteins as large as 900 residues. Therefore, we conclude that naturally knotted proteins cannot attain their knot randomly in the unfolded state but must pay the cost of knotting through contacts along their folding landscape.

[1] Departamento de Bioquímica y Biología Molecular, Facultad de Ciencias Químicas y Farmacéuticas, Universidad de Chile, Santos Dumont 964, Independencia, Santiago 8380494, Chile. [2] Departamento Académico de Ciencias Exactas, Facultad de Ciencias y Filosofía, Universidad Peruana Cayetano Heredia, Av. Honorio Delgado 430, San Martin de Porras, Lima−31 15102, Peru. [3] Laboratorio de Moléculas Individuales, Facultad de Ciencias y Filosofía, Universidad Peruana Cayetano Heredia, Av. Honorio Delgado 430, San Martin de Porras, Lima-31 15102, Peru. [4] Department of Molecular and Cell Biology, Department of Physics and Department of Chemistry, Kavli Energy Nanoscience Institute, and Howard Hughes Medical Institute, University of California, Berkeley, CA 94720 USA. Bustamante A and Sotelo-Campos J contributed equally to this work. Correspondence and requests for materials should be addressed to C.B. (email: carlosjbustamante2@gmail.com) or to M.B. (email: mauricio.baez@ciq.uchile.cl)

In natural proteins, the information encoded in their amino acid sequences directs their folding and attainment of the native state following precise kinetic and thermodynamic principles. Interestingly, analysis of the protein data bank reveals that only a small fraction (about 1%[1, 2]) of all folded proteins form knots in their interior. This observation is somewhat surprising, given that knots are abundant in ensembles of generic equilibrated polymers[1, 3], where entanglements inevitably arise with increasing chain length and compactness[4–8]. One common explanation for this bias is that knotting is not easily reconciled with the kinetics of folding because chains would require several well-coordinated steps to form a knot in a precise conformation[9–11]. According to this view, most proteins could avoid knots because the kinetics of knotting is complex and slow[12–14] relative to that of folding. In this view, kinetic mechanisms encoded on a selected group of sequences play a central role in minimizing the entanglement of unknotted structures and guide the efficient formation of knots for that subset of chains.

Yet, a paucity of information exists today regarding the thermodynamic and kinetics principles that govern knot formation in globular proteins. Specifically: How are the folding path and the folding rate of these proteins affected by the knots? What prevents other proteins from undesired knotting? And what is the functional role, if any, of natural knots? Here, to address some of these questions, we determine the energetic cost of knotting of an unfolded polypeptide chain and its effect on the folding process.

Numerical simulations of extended random self-avoiding homopolymer models suggest that knots should be rare in extended configurations of proteins[4, 15, 16]. However, these predictions have not been tested experimentally, because it is difficult to detect the presence of knots among ensembles of non-native states. Moreover, many denatured protein states show various degrees of residual structure and flexibility unaccounted for in these calculations[17–19] and, denatured states of some naturally knotted proteins appear to have great difficulty to untie spontaneously in experiments performed in bulk[12, 20, 21]. In such cases, the application of direct approaches, like nuclear magnetic resonance[22] and small-angle X-ray scattering[23] have been inadequate to establish the unknotted and knotted populations in the denatured state. Recently, Ziegler et al.[13], have studied the presence or absence of a knot in the mechanically unfolded state of ubiquitin C-terminal hydrolase isoenzyme L1 (UCH-L1) by pulling its structure from different points with optical tweezers. However, the structural complexity and the presence of multiple intermediates prevented them from determining the effect of a knot on the thermodynamic stability of a protein and the cost of its spontaneous formation in the mechanically unfolded state[13]. We have developed an alternative experimental approach based on the mechanical denaturation of a small artificial protein that overcomes these limitations.

Phage 22 ARC repressor is a homodimer with two RHH motifs[24] whose single-chain version—dubbed Arc-L1-Arc[25, 26]—has been proposed as the first artificial protein containing a $3_1$ knot[27]. Using optical tweezers, we found that Arc-L1-Arc not only presents a $3_1$ knot but also populates an unknotted configuration in its native state. We show that the native unknotted and knotted states of Arc-L1-Arc are almost isoenergetic, whereas these structures present different free energies of stabilization with respect to their respective mechanically unfolded states. Therefore, most of the difference in protein stability is due to the presence of the $3_1$ knot trapped in the denatured state of the knotted form. The free energy cost to form a trefoil knot in the denatured state of Arc-L1-Arc is high ($5.8 \pm 1$ kcal mol$^{-1}$) and also independent of the protein sequence since this value is well predicted by Monte Carlo simulations of random chains[5, 6]. Application of random chain models to calculate the knotting cost for longer polypeptides predicts that knots are rare events in denatured states of proteins in general. Further analysis of unfolding and refolding force distributions indicates a moderate effect of a knot on the position of the transition state and unfolding barrier, together with a significant decrease of the distance from the denatured to the transition state. Altogether, we conclude that knots are avoided by their high cost of formation in unfolded chains and we surmise that naturally knotted proteins must have evolved specific folding pathways to pay the cost of knotting through contacts along their folding landscape.

## Results

### Mechanical unfolding of Arc-L1-Arc reveals its dual topology.

Models of Arc-L1-Arc predict that it should form a $3_1$ knot when the artificial L1 loop passes over the C-terminal helix of the last RHH motif (Fig. 1a, loop in turquoise) and an unknotted conformation otherwise (Fig. 1a, L1 loop in magenta). The scoring energy of 6000 structures generated by the Rosetta software[28] indicates that the lowest energies of the unknotted and knotted conformations of Arc-L1-Arc are almost isoenergetic (Fig. 1b). Therefore, there is a reasonable chance to pick up either one of the two conformations by mechanically pulling on the folded protein near the N- and C-termini in an equilibrated ensemble. Two DNA handles[29] were attached to cysteine residues placed near the C- and N-terminal ends of Arc-L1-Arc trapping each molecule in only one knotted or unknotted conformation (Fig. 1c, left and right panels, respectively). We took advantage of this feature to compare the energy associated with the unfolding of a knotted and unknotted protein and to determine the energetic cost to form a knot in the unfolded state.

Figure 2a shows the force–extension curves of two Arc-L1-Arc molecules (upper and bottom panels) stretched and relaxed several times at 100 nm s$^{-1}$. Both molecules display single folding/unfolding transitions with unimodal force distributions, indicating that Arc-L1-Arc obeys a two-state unfolding mechanism. However, the molecule in the upper panel unfolds and refolds at lower forces ($9.7 \pm 1.9$ and $4.6 \pm 0.4$ pN, respectively) than that in the lower panel ($16.2 \pm 3.5$ and $7.0 \pm 0.4$ pN, respectively). Each tested molecule followed either one of these unfolding/refolding behaviors (Supplementary Fig. 1), resulting in a bimodal distribution for the entire data set of 32 Arc-L1-Arc molecules (Fig. 2b). Cluster analysis of unfolding and refolding peers (Supplementary Fig. 2a and Supplementary Note 1) confirms the conclusion arrived at by visual inspection: the heterogeneous behavior of unfolding and refolding forces arises from two types of Arc-L1-Arc molecules that do not interconvert in the presence of the DNA handles. Table 1 shows the average forces for each type of molecule obtained from two histograms generated by pooling the data of both types separately. Force–extension curves of Arc-L1-Arc molecules that undergo folding/unfolding transitions at high and low forces (Fig. 3a) yield contour lengths ($L_c$) of $36.9 \pm 2.8$ nm (Fig. 3b) and $40.2 \pm 3.8$ nm (Fig. 3c), respectively. The latter value is similar to that calculated for the unknotted conformation of Arc-L1-Arc (42 nm), while a tight $3_1$ knot is predicted to decrease a chain's $L_c$ by $4.7 \pm 0.4$ nm[30, 31].

The heterogeneous behavior of Arc-L1-Arc was not detected when pulling a non-circular permutant of it, pARC. This molecule retains the architecture of Arc-L1-Arc without the linker L1, making it impossible to form a knot[32] (Supplementary Fig. 3). The scatter plot of its unfolding and refolding forces coincides with that of Arc-L1-Arc molecules having low unfolding and refolding forces (Fig. 2b and Supplementary Fig. 2b). Also, pARC's $L_c$ ($37.1 \pm 3.8$ nm) agreed well with its theoretical value for the fully unfolded protein (36.05 nm, Supplementary Fig. 4a, b). Together, these results indicate that

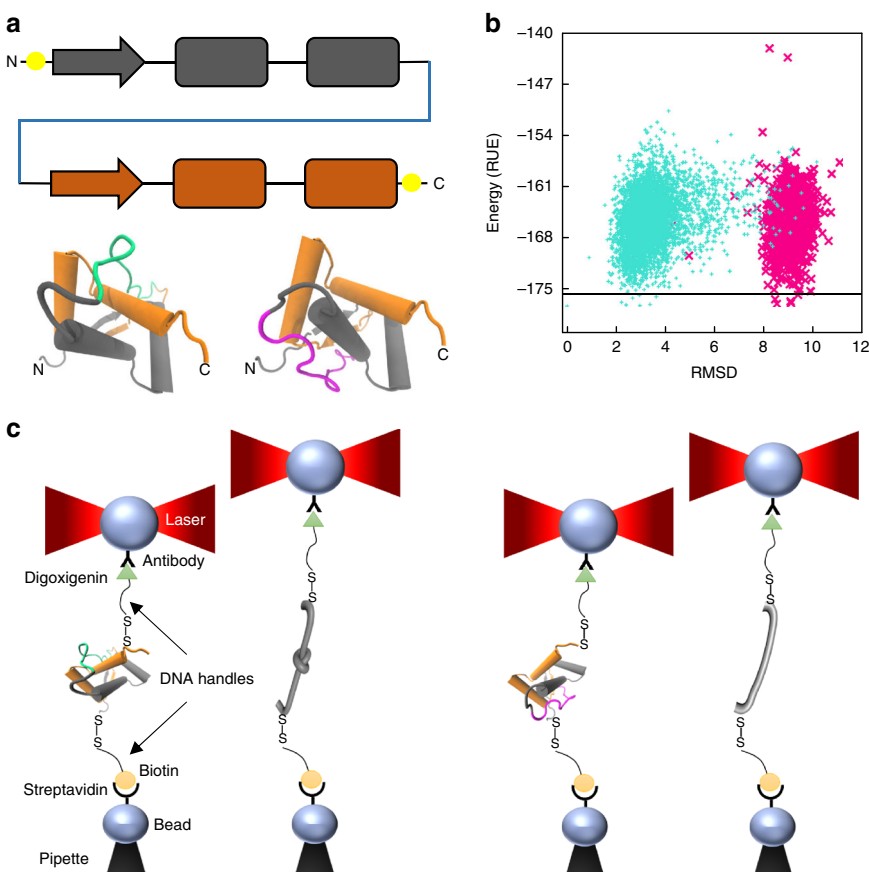

**Fig. 1** Dual conformation of Arc-L1-Arc. **a** Upper panel, schematic representation of the single chain Arc-L1-Arc molecule. RHH motifs are depicted in orange and gray. The L1 linker is depicted in blue. For the optical tweezer experiments, cysteine residues were introduced at positions 3 and 121 (yellow dots). Lower panel, molecular models of Arc-L1-Arc show that the protein can adopt either knotted (turquoise) or unknotted (magenta) topologies depending on the conformation of the L1 linker. **b** Energy vs. RMSD plots for loop L1 of Arc-L1-Arc determined by Rosetta. The turquoise and magenta dots represent 6000 loop conformations forming a $3_1$-knotted or unknotted structures of Arc-L1-Arc, respectively. The black horizontal line indicates 2 Rosetta Energy Units (REU) over the lowest energy structure predicted by Rosetta and represents a threshold to highlight that the 15 most probable structures of Arc-L1-Arc are either knotted or unknotted (2 REU is about 1 kcal mol$^{-1}$). **c** Optical tweezer experimental setup. If the protein is captured with the L1 linker in turquoise conformation (left), the protein forms a tight knot upon mechanical unfolding. Alternatively, if the L1 linker is in magenta conformation (right), the protein unfolds fully extended. Models of Arc-L1-Arc are based on the structure of phage P22 ARC (PDB ID: 1ARR)

the linker L1 in Arc-L1-Arc allows the protein to form either a knotted or an unknotted structure that are trapped and rendered non-interconvertible by placing the DNA handles.

**Calculation of the energy cost of polypeptide knot.** Does the formation of a knot confer thermodynamic besides mechanical stability to the molecule? To address this issue, we determined the free energy differences between the folded and unfolded states for the knotted and unknotted Arc-L1-Arc molecules ($\Delta G_{CFT}$). Since the folding and unfolding transitions occur irreversibly (away from equilibrium), we used the Crooks fluctuation theorem (CFT) to obtain these free energies from irreversible work distributions[33–36] of unknotted (Supplementary Fig. 5a) and knotted molecules (Supplementary Fig. 5b). The values of $\Delta G_{CFT}$ for folding of unknotted (Fig. 4a, upper equilibrium) and knotted molecules of Arc-L1-Arc (Fig. 4a, lower equilibrium) were $\Delta G_{CFT}$ (unknotted) $= 6.2 \pm 0.2$ kcal mol$^{-1}$ and $\Delta G_{CFT(knotted)} = 12.2 \pm 0.2$ kcal mol$^{-1}$, respectively, after correcting for the reversible work of stretching the unfolded chain in the pulling process (see Methods section). Folded and unfolded lifetimes plotted as a function of force intersect at the force where the molecule spends 50% of the time folded or unfolded ($F_{1/2}$; Supplementary Fig. 6, right panel); together with the change of extension between the folded and

unfolded state (Supplementary Fig. 6), $F_{1/2}$ can be used to obtain an independent value for the folding free-energy ($\Delta G_{F1/2}$)[37]. Figure 4a shows the values of $\Delta G_{F1/2}$ obtained for unknotted (upper equilibrium; $\Delta G_{F1/2}$ (unknotted) $= 5.4$ kcal mol$^{-1}$) and knotted (lower equilibrium; $\Delta G_{F1/2}$ (knotted) $= 12.3$ kcal mol$^{-1}$) states of Arc-L1-Arc. These values are consistent with those derived from the CFT analysis (see Table 1).

The extra thermodynamic stability conferred to the molecule by the knot cannot have its origin in differences between the native states of the knotted and unknotted Arc-L1-Arc, since the Rosetta scoring potential predicts them to be nearly isoenergetic (Fig. 1b, $\Delta G_N = G_{N,unknotted} - G_{N,knotted} = 0.5$ kcal mol$^{-1}$, see Methods section for calculation). A similar value is obtained ($\Delta G_N = 0.9$ kcal mol$^{-1}$) assuming that the 6 unknotted and 26 knotted folded molecules captured in the optical tweezers represent the equilibrium population between these native states. Therefore, the knotted protein gains an extra thermodynamic stability of about 6 kcal mol$^{-1}$ ($\Delta G_{CFT}$ (knotted)) because its denatured (knotted) reference state is destabilized relative to the denatured unknotted state by this amount. This quantity should be the energy cost to form a $3_1$ knot in the unfolded state of Arc-L1-Arc. Formally, the energy cost to form a $3_1$ knot in the unfolded state (Fig. 4a; $\Delta G_U$) can be obtained from the thermodynamic cycle connecting the native and denature

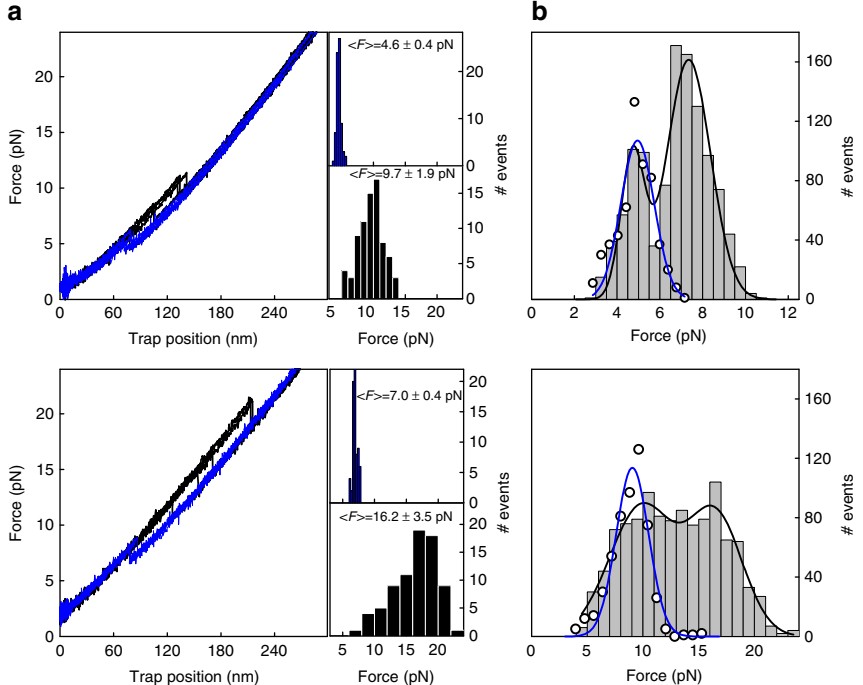

**Fig. 2** Two types of Arc-L1-Arc molecules. **a** Left. Force–extension curves of the unfolding (black) and refolding (blue) of two Arc-L1-Arc-DNA molecules (upper and bottom panels). Right. Force distributions of refolding (blue bars) and unfolding (black bars) events obtained from the force–extension curves of the same two molecules. The average force values and standard deviations are indicated in each graph. The force distribution histograms contain a total of 154 (upper) and 151 (bottom) transitions, respectively. **b** Force distribution of folding (upper, $n = 1108$) and unfolding (bottom, $n = 1074$) rips obtained upon stretching of 32 Arc-L1-Arc molecules (gray bar histograms). The continuous lines in black represent the fit to a sum of two Gaussian functions. Fitting parameters for refolding force distributions, $\mu_1 = 4.7$, $\mu_2 = 7.3$, $\sigma_1 = 0.5$, $\sigma_2 = 1$ and unfolding distribution: $\mu_1 = 10$, $\mu_2 = 16.5$, $\sigma_1 = 3$, $\sigma_2 = 2.5$. The force distribution for the refolding and unfolding of 12 pARC molecules is superimposed as white circles, and the continuous blue line represents its fit to a Gaussian function. Fitting parameters: $\mu_{\text{refolding}} = 5$, $\mu_{\text{unfolding}} = 9$, $\sigma_{\text{refolding}} = 0.8$, $\sigma_{\text{unfolding}} = 1.5$

conformations of knotted and unknotted of Arc-L1-Arc (Fig. 4a). Using the value of $\Delta G_N$ (left vertical equilibrium in Fig. 4a) and stability values determined experimentally for the unknotted and knotted conformations of Arc-L1-Arc, we calculate a free energy difference for the spontaneous formation of a $3_1$ knot in the unfolded state ($\Delta G_U$) of $5.8 \pm 1$ kcal mol$^{-1}$ (right vertical equilibrium in Fig. 4a). Thus formation of a $3_1$ knot incurs a large energy penalty for the unfolded polypeptide.

**Random model predicts the energy cost of polypeptide knot.** The value of $\Delta G_U$ does not have an ensemble counterpart to be compared to. We thus adapted a Monte Carlo method previously applied to the knotting of dsDNA molecules in diluted solutions (modeled as chains of rigid impenetrable cylinders of equal length and diameter)[5, 6] to determine the equilibrium fraction of $3_1$ knots in the unfolded state of Arc-L1-Arc. The fraction of knotted configurations depends on the persistence length of the chain and its effective diameter (Supplementary Methods and Supplementary Fig. 7) and was used to calculate the entropic free energy of formation of $3_1$ knots in unfolded chains ($\Delta G_S$; Eq. 7 in Supplementary Methods). This value was compared with the free energy cost of creating a $3_1$ knot in the denatured state of Arc-L1-Arc deduced from the thermodynamic cycle ($\Delta G_U = 5.8 \pm 1$ kcal mol$^{-1}$; Fig. 4a). A match is obtained for an effective chain diameter ($D$) of 0.49 nm for the unfolded state of Arc-L1-Arc (see graphical interpolation in Fig. 4b). This value is close to the effective diameter of 0.58 nm estimated by intrinsic viscosity measurements for denatured protein chains[38]; moreover, the average radius of gyration of the simulated configurations ($3.5 \pm 0.06$ nm for an effective diameter of 0.49 nm, Supplementary Fig. 8) is very close to the $\sim 3.3$–3.5 nm obtained experimentally

for denatured states of similar size[17, 39]. Comparable $\Delta G_U$ and radius of gyration values are predicted by simulations of flexible open chains made up of beads (diameter = 0.4–0.5 nm) with the length of the Arc-L1-Arc repressor (100 beads)[40]. The correspondence between experiments and random model computations indicates that $\Delta G_U$ is largely of entropic origin and, therefore, independent of protein sequence.

Experiments have shown that some average properties, such as the radius of gyration, scale with the protein length and are independent of the sequence, as predicted by random chain models[41]. Our results suggest that the knotting probability of denatured proteins is another such property. Thus it should be possible to predict the knotting probability of denatured proteins as a function of the chain length. In Fig. 4c, we expand the range of our calculations to chains of 370 residues ($\sim 100$ Kuhn segments of length 1.4 nm or $\sim 3.7$ amino acids each and of effective diameter 0.49 nm). These calculations predict a value of $\Delta G_U$ of 4.4 kcal mol$^{-1}$ for chains of 100 Kuhns. The slow decay with chain length observed for $\Delta G_U$ indicates that knots are rare events for natural proteins (up to 250 Kuhns or 900 residues). This conclusion is also supported by simulations performed for closed chains. These simulations predict that the knotting probability becomes important ($\Delta G_U$ approaches zero) only for chain lengths of about 10,000 Kuhn segments[42]. The propensity to form a knot should be even lower since these simulations were performed using a chain diameter of 0.28 nm, smaller than that derived here (0.49 nm, Fig. 4b).

**A knot alters the folding kinetics of Arc-L1-Arc.** Using the Bell[43] and Dudko–Hummer–Szabo[44] models, we extracted kinetic information for the unfolding/refolding processes

**Table 1 Thermodynamic and mechanical properties of Arc-L1-Arc and pARC**

| Molecule | | Unfolding force (pN) | Refolding force (pN) | $L_c$ (nm) | $\Delta G_{CFT}$ (kcal mol$^{-1}$) | [a]$\Delta G_{F1/2}$ (kcal mol$^{-1}$) |
|---|---|---|---|---|---|---|
| Arc-L1-Arc | Low force (magenta molecules) | 8.2 ± 2.7 | 4.7 ± 0.5 | 40.2 ± 3.8 | 6.2 ± 0.2 | 5.4 ± 0.7 |
| | High force (turquoise molecules) | 14.7 ± 3.9 | 7.3 ± 1.0 | 36.9 ± 2.8 | 12.2 ± 0.2 | 12.3 ± 2.1 |
| pARC | | 9.0 ± 1.5 | 5.0 ± 1.0 | 37.1 ± 3.8 | 8.1 ± 0.1 | 8.3 ± 1.2 |

All values expressed as average ± SD. In the case of Arc-L1-Arc the molecules were divided in two types, low ($n = 6$) or high ($n = 26$) force. The force values were obtained from histograms generated by pooling the data of both types separately (Supplementary Fig. 6)
[a]Free energy obtained at force at which the unfolding and folding lifetimes are equal ($F_{1/2}$) times the change of extension between folded and unfolded states corrected by the stretching force of the unfolded state

(Supplementary Fig. 6, left panel). This analysis indicated that the knot in the native state of Arc-L1-Arc has a moderate effect on the unfolding barrier. The distances between the folded and transition states for the knotted and the unknotted proteins do not differ significantly ($\Delta x_u^{\ddagger} = 1.8–1.2$ nm, for the knotted; and $\Delta x_u^{\ddagger} = 3.2–1.6$ nm, for the unknotted) and their corresponding unfolding rates at zero force are also similar (Supplementary Table 1). These results are to be expected since the L1 loop creates a superficial knot that is not involved in the formation of the hydrophobic core of Arc-L1-Arc (Fig. 1a). In contrast, the widths of the refolding force distributions of the knotted and unknotted configurations display significant differences (Supplementary Fig. 6, upper and middle panels). The Dudko–Hummer–Szabo function describing the refolding force distributions[45] does not yield a reliable fitting in either case. Therefore, we sought to determine what modifications in the kinetic parameters ($\Delta x^{\neq}$, $k_f$, $\Delta G^{\neq}$) reproduced the differences observed in the refolding force distributions between knotted and unknotted chains (Supplementary Fig. 6, upper and middle panels). Simulations revealed that a decrease of ~4 nm in the distance from the unfolded to the transition state ($\Delta x^{\neq}$) of the knotted protein—compared to its unknotted counterpart—is necessary to obtain the distributions of forces determined experimentally (Supplementary Fig. 6, upper and middle panels, Supplementary Table 2). Since the positions of the unfolding barriers are virtually unchanged by the knot (Supplementary Table 1), this decrease points to a displacement of the position of the unfolded state minimum rather than to a movement of the transition state due to the presence of the knot.

## Discussion

The success of random chain models in predicting correctly the knotting probability of a protein has implications about how proteins avoid knots. Experiments[20, 21] and simulations with polymer lattices[46] have reported that knots are persistent topologies after denaturing naturally knotted proteins. Consequently, the large energetic penalty to form knots in the denatured state of proteins seems a natural way for unknotted proteins to avoid such topological traps. On the other hand, several studies indicate that knots in the native states of globular proteins are much less frequent than predicted by random chain models[47, 48]. Therefore, additional factors such as kinetic mechanisms encoded on sequences[49], the emergence of secondary structures in compacts states[47], etc., must play a role in reducing the frequency of knots observed among proteins.

The knotting cost calculated here has important consequences for the understanding of the folding of naturally knotted proteins. First, once the native knotted state has been attained, it finds itself "stabilized" relative to its knotted denatured state by an additional barrier of 6 kcal mol$^{-1}$. This added stability may explain the conservation of knots in some protein families. Knots observed in the denatured states of naturally knotted proteins, like α/β methyltransferases, are resilient structures difficult to untie[20, 21]. Kinetic trapping of these knots in the denatured state should

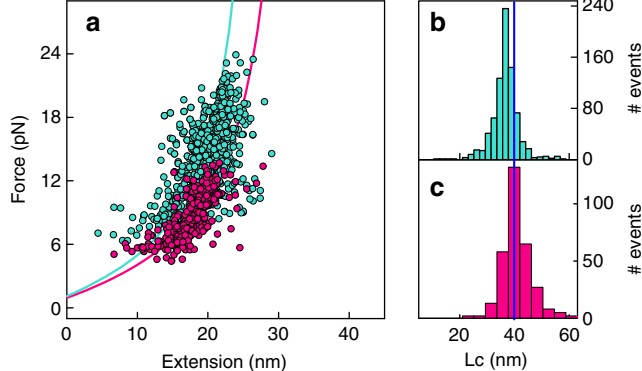

**Fig. 3** Force extension events for Arc-L1-Arc. **a** Plot of unfolding force vs. resulting extension events for molecules that unfold and refold at high forces (802 events, turquoise) and at low forces (287 events, magenta). The magenta and turquoise lines represent the WLC function calculated using $L_c$ of 40.1 and 37 nm, respectively, and a persistent length of 0.65 nm. Right. Contour length distribution for Arc-L1-Arc molecules that unfold/ refold at high **b** and low **c** forces. The blue line indicates the average $L_c$ obtained for Arc-L1-Arc molecules than unfolds/refolds at low forces. Note that the average contour length for molecules that unfold/refold at higher force (36.9 ± 2.8 nm) is significantly shorter ($p < 0.01$) than that for molecules that unfold/refold at lower force (40.2 ± 3.8 nm)

confer an effective thermodynamic stabilization to the proteins if their unknotted denatured states are not accessible in their biological time scale, as has been proposed[12]. Second, for proteins in which the knot is not superficial as in Arc-L1-Arc—but internal and integral to the folded structure—the high cost of knot formation in the denatured state implies that they must find a different kinetic path to attain the folded knotted state. This path, depicted schematically along the diagonal on Fig. 4a, corresponds to routes in which the price of knotting (5.8 ± 1 kcal mol$^{-1}$) is paid along the way through the simultaneous formation of side chain contacts as the protein collapses toward the folded knotted state[9, 12]. For example, it has been suggested that preordered intermediates would help to create the correct disposition of a loop for further chain threading[9, 10]. Such paths must have evolved among these proteins to ensure the efficient attainment of the knot during their folding. Finally, in some cases, such as the naturally knotted protein UCH-L1, the slower folding from unknotted unfolded states[13] suggests that some of this cost may not be paid entirely through folding contacts.

## Methods

**Modeling of dual conformations of Arc-L1-Arc.** The $3_1$ knot of Arc-L1-Arc has been inferred by comparison with the X-ray structure of the naturally knotted homologue VirC2 of *Agrobacterium tumefaciens*[27, 50]. VirC2 displays two RHH motifs connected by a loop that covers the C-terminal helix of the last RHH creating a $3_1$ knot in the polypeptide chain. Reconstruction of the L1 loop of Arc-

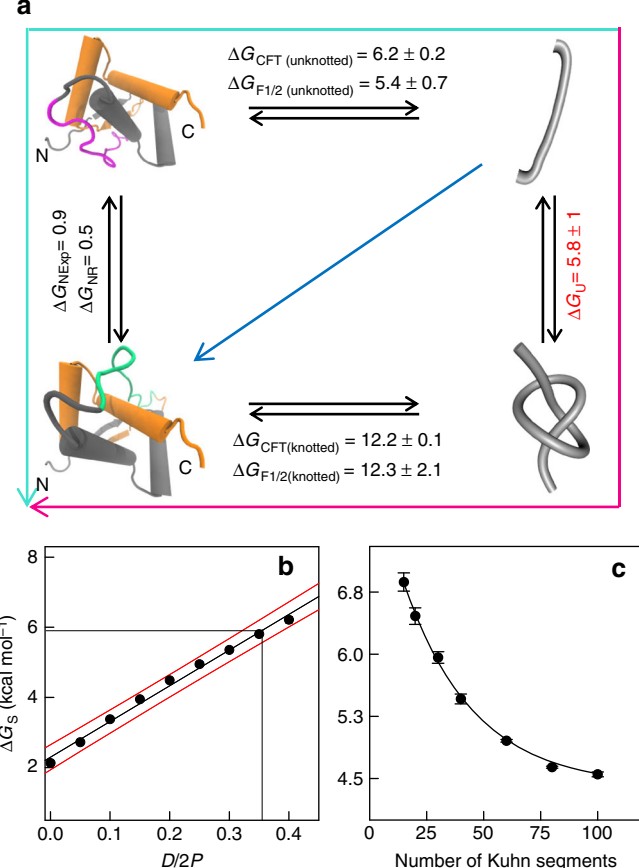

**Fig. 4** The energy cost of a knot in the denatured state of Arc-L1-Arc. **a** Thermodynamic cycle to display the interconversion reactions and free energies differences between the native and unfolded state of Arc-L1-Arc. The upper horizontal equilibrium depicts the reversible unfolding reaction between the native and denatured states of Arc-L1-Arc both in unknotted conformations. Likewise, the lower horizontal equilibrium represents the reversible unfolding equilibria when both states, native and unfolded, are forming a $3_1$ knot. The free energy differences for the unfolding reaction without ($\Delta G_{CFT\ (unknotted)}$, $\Delta G_{F1/2\ (unknotted)}$) or with a $3_1$ knot ($\Delta G_{CFT\ (knotted)}$, $\Delta G_{F1/2\ (knotted)}$) were calculated using Crooks fluctuation theorem ($\Delta G_{CFT}$) and the work at medium force ($\Delta G_{F1/2}$). The cycle is enclosed by the free energy differences of knotting in the native sate (left reaction, $\Delta G_N$) and knotting in the unfolded state (right reaction, $\Delta G_U$). The value of $\Delta G_U$ is determined once $\Delta G_N$ and the free energy differences calculated for the unfolding reactions of Arc-L1-Arc are quantified experimentally. **b** The entropic energy cost to form a $3_1$ knot ($\Delta G_S$) as a function of the relative chain diameter ($D/2P$). The innermost (central) curve corresponds to the best linear fit. The outer band (delimited by the red lines) depicts the 95% confidence interval. Interpolation for $\Delta G_U = 5.8$ kcal mol$^{-1}$ predicts a $D/2P$ value of 0.35, corresponding to $D = 0.49$ nm. **c** Entropic energy cost to form a $3_1$ knot ($\Delta G_S$) as a function of the chain length expressed in terms of the number of Kuhn segments ($N$) determined for chains with an effective diameter of 0.49 nm. The curve for all knots (with 10 or fewer crossings) is indistinguishable from the curve for $3_1$ knots. $N = 100$ represents a chain of about 370 residues. The line indicated in the figure corresponds to exponential fit valid only for the interval of chain length explored

L1-Arc create a $3_1$ knot as seen in VircC2. The presence of the $3_1$ knot in Arc-L1-Arc is dependent on the loop position. In order to explore the conformation of the L1 linker in the Arc-L1-Arc construct, a loop modeling prediction was applied using the Rosetta software[28]. First, a local all-atom search algorithm (FastRelax) was applied to the whole Arc dimer (PDB ID: 1ARR) in order to minimize its structure according to Rosetta's energy function[51]. This search gave us information about the structural flexibility of the anchorage residues for the L1 linker, which

made it possible to define the region of the protein to be conformationally explored. Six residues before (i.e., EGRIGA) and six residues after (i.e., MKGMSK) the inserted L1 linker (GGGSGGGTGGGSGGG) were included in the loop definition, totaling 27 residues. 6000 ab initio reconstructions of the loop region were produced by the KIC loop modeling algorithm[52], which were then topologically classified into knotted and unknotted structures. This assortment was carried out using the knot detection algorithm included into Rosetta's framework[53]. The root-mean-square deviation of every structure was calculated relative to the lowest energy model produced and plotted against their Rosetta score value in order to analyze the relative energy profile of the Arc-L1-Arc topological conformations.

The probability of obtaining a knotted or unknotted conformation for the structures modeled by Rosetta was calculated according to a Boltzmann distribution. First, the probability of each unknotted ($P_{unknotted}$) or knotted ($P_{knotted}$) conformation was calculated as:

$$P_i = \frac{e^{-(G_i/k_B T)}}{\sum_j^N e^{-(G_j/k_B T)}}$$

where $N$ is the total number of structures classified as knotted or unknotted, $i$ is the ith conformation, $G_i$ is the Rosetta energy obtained for conformation $i$, $k_B$ is the Boltzmann constant, and $T$ is the absolute temperature. The temperature term $k_B T$ was set to 1 Rosetta Energy Unit (REU). 1 REU corresponds to about 0.57 kcal mol$^{-1}$ based on comparative studies between experimentally determined $\Delta\Delta G$s and the REU predicted by Rosetta upon punctual mutations of proteins[54]. The free energy difference between knotted and unknotted structures ($\Delta G_N = G_{unknotted} - G_{knotted}$) was estimated by $\Delta G_N = -RT \ln(P_{unknotted})/(P_{knotted})$, where R is the ideal gas constant.

**Purification and chemical modification of proteins.** Genes encoding Arc-L1-Arc G3C/G121C and pARC G3C/G105C mutants were synthesized and purchased from Gen Script (Piscataway, NJ, USA, Supplementary Table 3). The proteins were expressed and purified as described previously by Robinson et al., 1996[25] except that a cationic exchange column was replaced by a step of size exclusion chromatography. *Escherichia coli* BL21 (DE3) cells (Thermo Fisher) containing the overexpressed proteins were harvested and lysed in 5 M GdnHCl, 10 mM Tris-HCl pH 8.0, 50 mM Phosphate, and 40 mM Imidazole. The lysate was loaded onto a HisTrap FF Crude (GE Healthcare) Ni$^{2+}$ affinity column and eluted with 5 M GdnHCl plus 0.2 M acetic acid. The eluate was reduced with 50 mM dithiothreitol (DTT) overnight. The reduced protein was refolded into a gel filtration column (Superdex 75/300 GL) equilibrated in 25 mM Tris, pH 7.6, 0.2 M KCl, and 0.2 mM EDTA. Samples eluted from the size exclusion column were immediately incubated with a 50-fold molar excess of 2,2'-Dithiodipyridine (DTDP) for 2 h at room temperature. The excess of DTDP was removed using two Micro Bio-Spin columns (Bio-Rad) equilibrated with 25 mM Tris pH 7.6, 0.2 M KCl, 0.1 mM EDTA, and 0.001% Tween-20. The DTDP–protein reaction was monitored spectro-photometrically at 343 nm observing the release of pyridine-2-thione ($\varepsilon_{343nm} = 7060$ M$^{-1}$cm$^{-1}$). The stoichiometry of DTPD modification was determined as the ratio between the pyridine-2-thione and the protein concentrations as described by Cecconi et al.[29].

**Attachment of DNA handles to proteins.** The 558 bp DNA handles were synthesized by PCR reaction using biotinilated, digoxigenin, and SH-modified oligos (Supplementary Table 4) as described by Cecconi et al. 2008[29]. A 30–40 μM SH-DNA handle solution in 10 mM Tris pH 8.5 was reduced with 10 mM DTT at room temperature for 2 h and then buffer exchanged into 25 mM Tris pH 7.6, 0.2 M KCl, 0.1 mM EDTA, and 0.001% Tween-20 using two Micro Bio-Spin columns (Bio-Rad). The reduced DNA solution was immediately mixed with the DTDP-activated proteins in molar ratio of 8:1 (DNA:Protein) during 6 h at room temperature following by 12 h at 4 °C. The DNA–protein complexes were purified in batch using Ni-NTA resin (QIAGEN). The unbound DNA was removed washing eight times with 25 mM Tris pH 7.6 and 0.2 M KCl. The complexes were eluted using the former buffer supplemented with 1 M imidazole. Once purified, the DNA–protein complex solution was supplemented with 0.1 mM EDTA and 50% Glycerol for storage at −80 °C.

**Optical tweezer experiments and data analysis.** Experiments were performed using a MiniTweezers device[55]. The protein was pulled at constant speed of 100 nm s$^{-1}$ between 2 and 30 pN[35, 56]. The molecules analyzed were those that displayed DNA overstretching transitions. Force–extension trajectories were analyzed using a Matlab program developed in Bustamante's laboratory.

**Contour length calculation.** The molecular extension change ($\Delta x_{exp}$) and the folding ($F_R$) and unfolding ($F_U$) forces of the stretched proteins was determined for each force–extension curve. In order to calculate the experimental contour length ($L_{c\ exp}$) the $\Delta x_{exp}$ data were compared with the molecular extension change predicted by the WLC model ($\Delta x_{theoretical}$) for each observed force. To calculate $\Delta x_{theoretical}$ values, the theoretical contour length ($L_{c\ theoretical}$) expected for the full Arc-L1-Arc (42 nm, 120 residues × 0.35 nm per aa) or pARC (36.05 nm, 103

residues × 0.35 nm per aa) and a persistence length of 0.65 nm were used as parameters for the WLC model. In this way, a $\Delta x_{\text{theoretical}}$ value was determined for each force. The $L_{c\ \text{exp}}$ distribution was calculated as follows:

$$L_{c\ \text{exp}} = \left(\frac{\Delta x_{\text{exp}}}{\Delta x_{\text{theoretical}}}\right) * L_{c\ \text{theoretical}}$$

**Calculation of reversible work of unfolding**. The work observed for each unfolding/refolding transition ($W_{\text{obs}}$) was calculated integrating the area of a rectangle delimited by the unfolding force on both sides of the transition. In this method, the stretching work of the DNA handles is automatically canceled[35]. In order to obtain the unfolding/refolding works in absence of forces ($W^{F=0}$), the work done to stretch the unfolded state ($W_s$) was subtracted from the work done to unfold mechanically the molecule ($W_{\text{obs}}$)[34]:

$$W^{F=0} = W_{\text{obs}} - W_s$$

The value of $W_s$ was determined by integration of the WLC between zero and each unfolding and refolding force of the protein. In this way, each value of $W_{\text{obs}}$ was corrected by the corresponding stretching work of the unfolded state. In the case of Arc-L1-Arc, the integration of WLC was obtained using a persistence length of 0.65 nm and a contour length of 42 or 37 nm depending on the type of molecule analyzed—those displaying low average unfolding/refolding forces (unknotted) or those displaying high average unfolding/refolding force (knotted)—respectively. For pARC, $W_s$ calculations were done using a 36.05 nm $L_c$ and a persistence length of 0.65 nm. $W_s$ calculations were done using a Matlab script, kindly provided to us by Jessie Dill and modified in order to calculate various $W_s$ at once. The distributions of $W^{F=0}$ were adjusted to a Gaussian function and the $\Delta G°$ was calculated as the point of intersection between the folding and unfolding probability distributions[33]. The $\Delta G_{F1/2}$ was calculated as $F_{1/2}$ times $\Delta x_{(F1/2)}$, corrected by the stretching work of the unfolded state ($W_s$). $F_{1/2}$ was determined as the point of intersection of the force dependent lifetimes of the folding/unfolding process and $\Delta x_{(F1/2)}$ by using the $L_c$ of the unknotted and knotted Arc-L1-Arc.

**Data analysis**. The analysis was done using SigmaPlot 10. Histograms were generated using the automatic binning option. The algorithm calculates the number of bins for representation based upon the number of data points according to:

$$\text{No. of bins} = 3 + \frac{\log(N)}{\log(2)}$$

where $N$ is the number of data points.

Fitting procedures to force distributions were performed using the dynamic fit option of SigmaPlot 10.

**Cluster analysis**. Each folding force ($F_R$) was paired with its corresponding unfolding force ($F_U$). Each ($F_R$, $F_U$) pair (available at https://doi.org/10.6084/m9.figshare.5195710.v1)[57] was represented as a point on a two-dimensional plane. Clustering was done using the SPSS software by applying a Biphasic agglomerative hierarchical algorithm[58]. Similarity between data points was determined by calculating the distance between them in the plane. In this case, $A$ and $B$ are data points with coordinates ($F_R^A$, $F_U^A$) and ($F_R^B$, $F_U^B$), respectively.

Accordingly, the shorter the distance between two ($F_R$, $F_U$) pairs of points the higher the likelihood that these points will be clustered in the same class. The first step of the biphasic algorithm is a preclustering to form groups of points that are near each other. Once preclusters are formed, they are classified using a hierarchical clustering algorithm based on a centroid method. For the set of preclusters formed in the first step, a similarity matrix is computed by using the distance between preclusters' centroids. The matrix is scanned to identify the lower value representing the most similar preclusters. Those preclusters are joined and the similarity matrix is updated by the centroid's preclusters, replacing the joined elements. The process continues until only one element is left.

**Code availability**. The source code for the version of the computer program used in this study is available from the corresponding authors upon reasonable request.

**Data availability**. Data set of force, distance, and work: https://doi.org/10.6084/m9.figshare.5195710.v1. All other data supporting these findings are available from the corresponding authors upon reasonable request.

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

## Acknowledgements

This work was supported by Fondecyt 11110534 (to M.B.), 1151274 (to M.B.), 11130263 (to C.A.M.W.), Anillo 1107 (to M.B.), and Chile and Fondecyt 196–2013 (to D.G.G.) CONCYTEC, Perú. This research was partially supported by the super-computing infrastructure of the NLHPC (ECM-02) at Universidad de Chile. A.B. was supported by Conicyt master fellowship 22121199. We gratefully acknowledge Professor Alexander Vologodskii from New York University for helpful comments and advice. Travel for C.B. to and from Chile and Peru was also partially supported by the Howard Hughes Medical Institute, NIH grant R01GM032543, and the U.S. Department of Energy Office of Basic Energy Sciences Nanomachine Program under contract no. DE-AC02-05CH11231.

## Author contributions

Conceived and designed the experiments: M.B., C.B., D.G.G., A.B. Performed the experiments: M.B., A.B., J.S-C, M.F., C.A.M.W. Analyzed the data: A.B., M.F., C.A.M.W., M.B., J.S.-C. Wrote the paper: M.B., C.B., D.G.G., J.S.-C.

## Additional information

**Competing interests:** The authors declare no competing financial interests.

**Change history:** A correction to this article has been published and is linked from the HTML version of this paper.

