## [Peer Review file · Nature Communications]

Reviewers' comments:

Reviewer #1 (Remarks to the Author):

The manuscript of Bustamante et al presents a characterization of the free-energy cost of introducing a knot in a peptide chain.

The approach is based on a careful combination of single-molecule experiments and statistical-mechanical theory (Jarzynski's relations and Monte Carlo simulations). The authors show that a significant entropic barrier needs to be overcome to introduce a knot when the peptide chain is still largely unstructured.

Both the approach and the results are very interesting and hence I enthusiastically recommend publication of the article.

I recommend to address the following points in the revised manuscript:

- To my knowledge, the entropic cost of tying knots in open chains was first addressed by Tubiana et al, (Macromolecules 2013), who also reported on the dynamical mechanisms leading to the spontaneous knotting of chains of beads (i.e. without sequence-specific interactions). The free energy cost found here by Bustamante et al. appears comparable with those previously reported for such system. First, based on the considerations at lines 157-174, Table 1 and Figs. S6, one has that the unfolded Arc-L1-Arc about corresponds to a fully-flexible chain of 100 beads of diameter 0.4-0.5nm. For such system D'Adamo and Micheletti (Macromolecules 2015) report an equilibrium gyration radius of $7.46 \times 0.5 \text{ nm} = 3.4 \text{ nm}$, consistent with the one measured here, and a knotting probability of 2×10^{-6} , which corresponds to an entropic cost of 7.5Kcal/mol. This is not dissimilar from the range of values reported by the authors for ΔG_U , and gives a further independent support to it.

- I find it remarkable that the measured free energy cost of introducing a knot in the unfolded/denatured state is close to that inferred from general, aspecific polymer models. In some ways it had been suspected before that proteins in the unfolded/denatured states could behave as flexible homopolymers, but I believe this is the first time that it is (indirectly) shown to be so for knotting properties too. Do the authors believe this is result that can hold more in general, beyond this specific protein? This is an important point, because various considerations indicate that proteins are far less knotted than equivalent globular homopolymers (Luo and Grosberg, Plos Comput Biol 2006, Soler et al. J. Chem. Phys 2014, Wuest et al Phys Rev. Lett. 2015, Jackson et al. Curr Op. Struct Biol 2017). Those considerations have, by necessity, been restricted to native states. Showing that in the unfolded/denatured state proteins have the same knotting propensity of equivalent homopolymers would open important perspectives, particularly regarding when native or non-native interactions kick in to tilt the balance in favour of the knotted or unknotted native state.

Reviewer #2 (Remarks to the Author):

This paper reports on the energetic cost of knotting in unfolded polypeptide chains and the effect of the knot formation on the stability and folding kinetics of a protein. Mechanical denaturation by optical tweezers pulling of the artificial protein Arc-L1-Arc (known to present 31 knot) is presented as a study case. The findings are novel and noteworthy, shining new light on the understanding of the thermodynamic and kinetics of knot formation in proteins. Besides the interesting findings, I was particularly impressed by the project planning, starting from properly choosing the molecule, following with the experimental approach, computational techniques and concluding with worthwhile data interpretation. In my opinion this is an example of a well formulated scientific problem which, as in geometry, being well formulated it is half solved. Beautiful.

Altogether, considering the experimental and theoretical approaches, the results and the

conclusions of this manuscript I do recommend it for publication in Nature Communications, with minor revision addressing my comments underneath.

1. Analyzing the scoring energy of 6000 structures generated by Rosetta software, it is shown that the knotted and unknotted conformations are almost isoenergetic (Fig. 1B). The meaning of the criterium to choose the threshold (cutoff 2 REU ?) delimitating the lowest energy is not clear.

Please detail in the text or figure legend. Fig. 1C right panel presents a vertical black line without any meaning. Please remove or explain.

2. Using optical tweezers to pull the folded protein near the N- and C- termini, and measuring the force vs extension, the authors elegantly show that Arc-L1-Arc molecules can be of two types, each type displaying single folding/unfolding transitions with unimodal force distributions but with distinct unfolding/refolding forces between the two types. The pulling experiment is applied also to pARC, a non-circular permutant of Arc-L1-Arc protein, showing a monomodal behavior

corresponding to lower force, which is attributed to the absence of the linker L1 in pARC. The presence of the linker L1 is thus necessary but not sufficient for the knot formation. As regards Fig.2 and fitting the data with one and respectively two Gaussian functions: please specify the fitting parameters. The forces for unfolding data are more spread than for folding, both for the single molecule and the cluster of molecules. Is there any meaning associated to this? If yes, please comment.

3. Calculating the free energy differences between the folded and unfolded states for the knotted and unknotted Arc-L1-Arc protein molecules the authors find that the knotted protein gains an extra thermodynamic stability of about 6 kcal/mol which is attributed to the destabilization of the denatured knotted reference state with respect to the denatured unknotted state. Hence, this quantity is the energy cost to form a 31 knot in the unfolded state of Arc-L1-Arc. The free energy difference for the spontaneous formation of a 31 knot in the unfolded is also derived (5.8 kcal/mol, which is a large energy cost). Fig. 4 illustrates the thermodynamic characterization of the Arc-L1-Arc artificial protein folding. Although the idea of Fig. 4a is good, a better quality and more explanation in the legend and text would help the reader understand better. Fig. 4c shows the entropic energy vs the number of Kuhn segments (N) and a curve fitting which should support the conclusion that the energy cost for a knot formation is constant for $N > 40$. There are two comments here: 1) the lack of data from $N = 40$ to $N = 100$, so the fitting is unclear; 2) considering the fitting presented in the figure, the curve seems to be saturated for $N > 100$ and not for $N > 40$ (the energy difference between $N = 40$ and 100 is about 1 kcal/mol). Please add data for $40 < N < 100$ and explain the fitting and the saturation. Another comment for Fig. 4C regards the energy axis: using an offset value 4 kcal/mol the figure can be reduced, saving space.

Response to reviews for Manuscript NSMB-A36110

First, we wish to thank the reviewers for their thorough and useful comments on the manuscript. As described below, we have carefully considered their suggestions and made pertinent changes to the manuscript and figures.

Our responses (in blue) are inserted directly below each individual comment.

In the revised manuscript general changes (syntax and grammar) are in green and specific editions in response to reviewer's comments are in blue.

General modifications.

In material and methods, the experimental procedures corresponding to subtitle "Equilibrium denaturation curves" were eliminated. These experiments are not included or described in this study.

In material and methods, the first paragraph corresponding to "Cluster analysis" was moved to "Supplementary Text".

Reviewer # 1

The manuscript of Bustamante et al. presents a characterization of the free-energy cost of introducing a knot in a peptide chain. The approach is based on a careful combination of single-molecule experiments and statistical-mechanical theory (Jarzynski's relations and Monte Carlo simulations). The authors show that a significant entropic barrier needs to be overcome to introduce a knot when the peptide chain is still largely unstructured. Both the approach and the results are very interesting and hence I enthusiastically recommend publication of the article.

I recommend to address the following points in the revised manuscript:

- To my knowledge, the entropic cost of tying knots in open chains was first addressed by Tubiana et al, (Macromolecules 2013) who also reported on the dynamical mechanisms leading to the spontaneous knotting of chains of beads (i.e. without sequence-specific interactions). The free energy cost found here by Bustamante et al. appears comparable with those previously reported for such system. First, based on the considerations at lines 157-174, Table 1 and Figs. S6, one has that the unfolded Arc-L1-Arc about corresponds to a fully-flexible chain of 100 beads of diameter 0.4-0.5nm. For such system D'Adamo and Micheletti (Macromolecules 2015) report an equilibrium gyration radius of $7.46 \times 0.5 \text{ nm} = 3.4 \text{ nm}$, consistent with the one measured here, and a knotting probability of 2×10^{-6} , which corresponds to an entropic cost of 7.5Kcal/mol. This is not dissimilar from the range of values reported by the authors for ΔG_U , and gives a further independent support to it.

Response:

We thank the reviewer for this suggestion. In effect, the article of D'Adamo and Micheletti (Macromolecules 2015) also uses Monte Carlo simulations to estimate the knotting

probability of open chains of beads. We have included this observation into the revised manuscript to give independent support to our own simulations performed with open chains of cylinders (page 9, first paragraph).

-I find it remarkable that the measured free energy cost of introducing a knot in the unfolded/denatured state is close to that inferred from general, aspecific polymer models. In some ways it had been suspected before that proteins in the unfolded/denatured states could behave as flexible homopolymers, but I believe this is the first time that it is (indirectly) shown to be so for knotting properties too. Do the authors believe this is result that can hold more in general, beyond this specific protein?

This is an important point, because various considerations indicate that proteins are far less knotted than equivalent globular homopolymers (Lua and Grosberg, Plos Comput Biol 2006, Soler et al. J. Chem. Phys 2014, Wuest et al Phys Rev. Lett. 2015, Jackson et al. Curr Op. Struct Biol 2017).

Those considerations have, by necessity, been restricted to native states. Showing that in the unfolded/denatured state proteins have the same knotting propensity of equivalent homopolymers would open important perspectives, particularly regarding when native or non-native interactions kick in to tilt the balance in favour of the knotted or unknotted native state.

Response:

The reviewer raises a very interesting point. Indeed, we believe that the behavior observed in our experiments can be applied to other proteins. Random chain models have been used to predict average properties of proteins in good solvents like Urea or GdnHCl. For example, the radius of gyration of chemically denatured states of several proteins has been found to scale linearly with the chain length, as is expected by random polymers theories (Kohn et al., 2004). In this respect, the denatured state of Arc-L1-Arc shows physical properties expected of denatured states of polypeptide chains of similar size (Robinson and Sauer, 1996). Since our experimental estimate of the energy cost of knotting is well predicted by a random model, in principle, the knotting probabilities of denatured proteins of different lengths can be predicted by using such models. In this vein, we have computed such probabilities and used them to calculate the free energy of knotting as a function of the chain length (Figure 4C). Therefore, our results provide experimental support for an entropic origin of the knot formation in the denatured state of proteins. As indicated by the reviewer, this observation implies that additional factors such as specific interactions (Wust et al., 2015) and perhaps the early formation of secondary structure (Lua RC, 2006), must also play a role in making knots even rarer in the native state of proteins. Conversely, these same factors may also contribute to the guiding of knot formation in a reduced number of naturally knotted proteins. To further emphasize this important point raised by the reviewer, we include the arguments presented in this paragraph in the revised version of the manuscript (page 9, second paragraph and page 10, last paragraph).

Reviewer #2 (Remarks to the Author):

This paper reports on the energetic cost of knotting in unfolded polypeptide chains and the effect of the knot formation on the stability and folding kinetics of a protein. Mechanical denaturation by optical tweezers pulling of the artificial protein Arc-L1-Arc (known to present 31 knot) is presented as a study case. The findings are novel and noteworthy, shining new light on the understanding of the thermodynamic and kinetics of knot formation in proteins. Besides the interesting findings, I was particularly impressed by the project planning, starting from properly choosing the molecule, following with the experimental approach, computational techniques and concluding with worthwhile data interpretation. In my opinion this is an example of a well formulated scientific problem which, as in geometry, being well formulated it is half solved. Beautiful. Altogether, considering the experimental and theoretical approaches, the results and the conclusions of this manuscript I do recommend it for publication in Nature Communications, with minor revision addressing my comments underneath.

1. Analyzing the scoring energy of 6000 structures generated by Rosetta software, it is shown that the knotted and unknotted conformations are almost isoenergetic (Fig. 1B). The meaning of the criterium to choose the threshold (cutoff 2 REU ?) delimitating the lowest energy is not clear. Please detail in the text or figure legend. Fig. 1C right panel presents a vertical black line without any meaning. Please remove or explain.

Response:

The value of 2 REU corresponds to ~ 1 kcal/mol according to (Kellogg et al., 2011). Since 1 kcal/mol is close to the energy of the thermal bath at 298 K, we used this value to conclude that both the knotted and unknotted structures of Arc-L1-Arc predicted by Rosetta should be well populated at room temperature. The legend of Fig. 1C has been modified according to this description. In addition, we calculated the free energy difference using the weighted Boltzmann factors of the entire data set of energies and structures calculated by Rosetta which predicted a difference in free energy between the knotted and unknotted configurations of the folded state to be 0.5 kcal/mol.

The vertical black line was removed from the figure.

2. Using optical tweezers to pull the folded protein near the N- and C- termini, and measuring the force vs extension, the authors elegantly show that Arc-L1-Arc molecules can be of two types, each type displaying single folding/unfolding transitions with unimodal force distributions but with distinct unfolding/refolding forces between the two types. The pulling experiment is applied also to pARC, a non-circular permutant of Arc-L1-Arc protein, showing a monomodal behavior corresponding to lower force, which is attributed to the absence of the linker L1 in pARC. The presence of the linker L1 is thus necessary but not sufficient for the knot formation. As regards Fig.2 and fitting the data

with one and respectively two Gaussian functions: please specify the fitting parameters. The forces for unfolding data are more spread than for folding, both for the single molecule and the cluster of molecules. Is there any meaning associated to this? If yes, please comment.

Response:

The distributions of unfolding and refolding forces do indeed contain kinetic information about the mechanical folding and unfolding reactions, respectively. Some theoretical treatments, like the Dudko–Hummer–Szabo and Bell models, can be applied to fit the experimental force distributions and thus recover important kinetic parameters like rates of folding/unfolding and the transition state distances (Bell et al., 1984; Dudko et al., 2008; Pierser and Dudko, 2013). Specifically, in the case of the Bell model, the folding and unfolding distances to the transition state are proportional to the steepness of the logarithm of the rate of folding or unfolding as a function of the applied force, respectively. Equivalently, the Bell model predicts that the width of the distribution of unfolding and refolding forces is inversely proportional to distance to the transition state. The unfolding forces of Arc-L1-Arc denaturation are more spread than the distribution of forces at which it folds, thus showing a similar behavior to other proteins that have been studied by mechanical denaturation (for example see, (Alemayehu et al., 2016; Elms et al., 2012; Goldman et al., 2015; Shank et al., 2010)). In general, it has been found that the compact/native states of a protein are relatively fragile or “brittle”, often requiring just a small deformation, relative to the total distance between the native and the mechanically unfolded state, to reach the transition state. Conversely, ‘molten globule states’ are known to be much more “compliant”, that is, associated with larger distances to the transition state (Elms et al., 2012).

Additional information, such as the value of the free energy barrier, can be obtained applying more comprehensive models like the Dudko–Hummer–Szabo function (Dudko et al., 2008; Pierser and Dudko, 2013). In the initially submitted manuscript, we analyzed the force distribution (unfolding and refolding) for both knotted and unknotted configurations of Arc-L1-Arc in terms of the Dudko–Hummer–Szabo function to determine how the kinetics are altered by the presence of a knot (Supplementary Table 1 and 2, and Supplementary Fig. 6). As is indicated in the page 10 of the revised manuscript, the knot induces a displacement of 4 nm when the folding reaction begins from the denatured-knotted state of Arc-L1-Arc:

“Simulations revealed that a decrease of ~ 4 nm in the distance from the unfolded to the transition state (Δx^\ddagger) of the knotted protein—compared to its unknotted counterpart—is necessary to obtain the distributions of Δx^\ddagger determined experimentally (Supplementary Fig. 6 upper and middle panels, Supplementary table 2). Since the positions of the unfolding barriers are virtually unchanged by the knot (Supplementary Table 1), this decrease points to a displacement of the position of the unfolded state minimum rather than to a movement of the transition state due to the presence of the knot”.

The fitting parameters for one- and two-Gaussian fits are now indicated in the legend of Fig. 2

Calculating the free energy differences between the folded and unfolded states for the knotted and unknotted Arc-L1-Arc protein molecules the authors find that the knotted protein gains an extra thermodynamic stability of about 6 kcal/mol which is attributed to the destabilization of the denatured knotted reference state with respect to the denatured unknotted state. Hence, this quantity is the energy cost to form a 31 knot in the unfolded state of Arc-L1-Arc. The free energy difference for the spontaneous formation of a 31 knot in the unfolded is also derived (5.8 kcal/mol, which is a large energy cost). Fig. 4 illustrates the thermodynamic characterization of the Arc-L1-Arc artificial protein folding. Although the idea of Fig. 4a is good, a better quality and more explanation in the legend and text would help the reader understand better.

Response

We thank the reviewer for his/her suggestion. Figure 4A has been edited to improve its quality; its legend includes now a better explanation of the depicted transitions. Additionally, in the main text we have explained how the thermodynamic value of ΔG_u was obtained from the thermodynamic cycle (page 7 and 8).

Fig. 4C shows the entropic energy Vs. the number of Kuhn segments (N) and a curve fitting which should support the conclusion that the energy cost for a knot formation is constant for $N > 40$. There are two comments here: 1) the lack of data from $N = 40$ to $N = 100$, so the fitting is unclear; 2) considering the fitting presented in the figure, the curve seems to be saturated for $N > 100$ and not for $N > 40$ (the energy difference between $N = 40$ and 100 is about 1 kcal/mol). Please add data for $40 < N < 100$ and explain the fitting and the saturation. Another comment for Fig. 4C regards the energy axis: using an offset value 4 kcal/mol the figure can be reduced, saving space.

Response

We agree with the reviewer that the curve appears to be saturated for $N > 100$ rather than for $N > 40$. Therefore, we replaced the sentence of the abstract: “a value nearly constant for all proteins over 120 residues” with “The free energy cost is predicted to remain above 3 kcal mol⁻¹ for proteins as large as 900 residues”.

In figure 4C we used a single exponential decay function to describe the variation of the data. The best fit with this model required an offset of 4.4 kcal/mol for infinite chain lengths. However, the fitting of the free energy of knotting should be considered accurate for proteins constituted by up to 100 Kuhn segments (or 370 residues) each segment assumed to be 1.4 nm long or ~3.7 amino acids, whereas the saturation value should be considered (within the model) a good approximation for longer proteins. For example, from the computational work of Uehara and Deguchi 2015 on closed chains of cylinders of the

same length as ours but of diameter 0.28 nm, we can deduce that the knotting probability increases very slowly with chain length (Uehara and Deguchi, 2015), tending asymptotically to 1 for chain lengths well beyond 10000 Kuhn segments (where the probability is 0.5, i.e., $\Delta G_U = 0$). For proteins, whose diameter our work predicts it to be 0.49 nm (figure 4B), ΔG_U should approach 0 at lengths longer than 10000 Kuhns. Thus, far beyond the length of natural proteins ($\gg 250$ Kuhns or 900 residues) the plateau approximation is no longer valid because the knotting probability should approach one for chains of infinite length (Delbruck, 1962; Frisch and Wasserman, 1961). To be consistent with these considerations and to avoid misinterpretations, we have made appropriate changes in the text to describe Figure 4C as well as made explicit the limitations of our calculations (see last paragraph of the page 4 and second paragraph of the page 9). In any case, the analysis indicated above doesn't change the original conclusion on the infrequency of knots in proteins reached from our results.

As was suggested by the reviewer, the entropic cost of knotting (ΔG_U) for chains with 60 and 80 Kuhn segments was calculated to fill the gap between 40 $<N < 100$ (see the new Figure 4C). The curve in Figure 4 C was almost unchanged with the addition of the new values of ΔG_U . The offset of Figure 4C was changed from 0 to 4 kcal/mol, as suggested by the reviewer. Now Figures 4A, B and C are arranged into a single panel to fulfill editorial requirements.

References.

- Aleman, A., Rey-Serra, B., Frutos, S., Cecconi, C., and Ritort, F. (2016). Mechanical Folding and Unfolding of Protein Barnase at the Single-Molecule Level. *Biophys J* 110, 63-74.
- Bell, G.I., Dembo, M., and Bongrand, P. (1984). Cell adhesion. Competition between nonspecific repulsion and specific bonding. *Biophys J* 45, 1051-1064.
- Delbruck, M. (1962). Mathematical problems in the biological sciences. In *Proc Symp Appl Math*, p. 55.
- Dudko, O.K., Hummer, G., and Szabo, A. (2008). Theory, analysis, and interpretation of single-molecule force spectroscopy experiments. *Proc Natl Acad Sci USA* 105, 15755-15760.
- Elms, P.J., Chodera, J.D., Bustamante, C., and Marqusee, S. (2012). The molten globule state is unusually deformable under mechanical force. *Proc Natl Acad Sci USA* 109, 3796-3801.
- Frisch, H.L., and Wasserman, E. (1961). Chemical topology1. *J Am Chem Soc* 83, 3789-3795.

Goldman, D.H., Kaiser, C.M., Milin, A., Righini, M., Tinoco, I., Jr., and Bustamante, C. (2015). Ribosome. Mechanical force releases nascent chain-mediated ribosome arrest in vitro and in vivo. *Science* *348*, 457-460.

Kellogg, E.H., Leaver-Fay, A., and Baker, D. (2011). Role of conformational sampling in computing mutation-induced changes in protein structure and stability. *Proteins* *79*, 830-838.

Kohn, J.E., Millett, I.S., Jacob, J., Zagrovic, B., Dillon, T.M., Cingel, N., Dothager, R.S., Seifert, S., Thiyagarajan, P., Sosnick, T.R., *et al.* (2004). Random-coil behavior and the dimensions of chemically unfolded proteins. *Proc Natl Acad Sci USA* *101*, 12491-12496.

Lua RC, G.A. (2006). Statistics of Knots, Geometry of Conformations, and Evolution of Proteins. *PLOS Comput Biol* *2*.

Pierse, C.A., and Dudko, O.K. (2013). Kinetics and energetics of biomolecular folding and binding. *Biophys J* *105*, L19-22.

Robinson, C.R., and Sauer, R.T. (1996). Equilibrium stability and sub-millisecond refolding of a designed single-chain Arc repressor. *Biochemistry* *35*, 13878-13884.

Shank, E.A., Cecconi, C., Dill, J.W., Marqusee, S., and Bustamante, C. (2010). The folding cooperativity of a protein is controlled by its chain topology. *Nature* *465*, 637-640.

Uehara, E., and Deguchi, T. (2015). Characteristic length of the knotting probability revisited. *Journal of physics. Condensed matter : an Institute of Physics journal* *27*, 354104.
Wust, T., Reith, D., and Virnau, P. (2015). Sequence determines degree of knottedness in a coarse-grained protein model. *Phys Rev Lett* *114*, 028102.

REVIEWERS' COMMENTS:

Reviewer #1 (Remarks to the Author):

The authors have adequately addressed my previous comments. I am very happy to recommend publication of the manuscript.

Reviewer #2 (Remarks to the Author):

The authors exhaustively answered all the comments. Thanks. Now I consider the manuscript ready for publication.